# COVID-19–related perceptions, context and attitudes of adults with chronic conditions: Results from a cross-sectional survey nested in the ComPaRe e-cohort

Viet-Thi Tran[1,2]*, Philippe Ravaud[1,2,3]

**1** Université de Paris, CRESS, INSERM, INRA, Paris, France, **2** Centre d'Epidémiologie Clinique, AP-HP, Hôpital Hôtel-Dieu, Paris, France, **3** Department of Epidemiology, Columbia University Mailman School of Public Health, New York, New York, United States of America

* thi.tran-viet@aphp.fr

## Abstract

### Background

To avoid a surge of demand on the healthcare system due to the COVID-19 pandemic, we must reduce transmission to individuals with chronic conditions who are at risk of severe illness with COVID-19. We aimed at understanding the perceptions, context and attitudes of individuals with chronic conditions during the COVID-19 pandemic to clarify their potential risk of infection.

### Methods

A cross-sectional survey was nested in ComPaRe, an e-cohort of adults with chronic conditions, in France. It assessed participants' perception of their risk of severe illness with COVID-19; their context (i.e., work, household, contacts with external people); and their attitudes in situations involving frequent or occasional contacts with symptomatic or asymptomatic people. Data were collected from March 23 to April 2, 2020, during the lockdown in France. Analyses were weighted to represent the demographic characteristics of French patients with chronic conditions. The subgroup of participants at high risk according to the recommendations of the French High Council for Public Health was examined.

### Results

Among the 7169 recruited participants, 63% patients felt at risk because of severe illness. About one quarter (23.7%) were at risk of infection because they worked outside home, had a household member working outside home or had regular visits from external contacts. Less than 20% participants refused contact with symptomatic people and <20% used masks when in contact with asymptomatic people. Among patients considered at high risk according to the recommendations of the French High Council for Public Health, 20% did not feel at risk, which led to incautious attitudes.

**Data Availability Statement:** Data from this study cannot be shared publicly because they are part of the ComPaRe e-cohort. As specified in their

consent on inclusion in the cohort, all data collected may be shared and reused but solely by academic researchers and after acceptance of their study protocol by the scientific committee of the ComPaRe e-cohort. The procedure is described in detail on the ComPaRe e-cohort website (www. compare.aphp.fr) and was validated by the ethics committee (CPP 1 d'Ile de France) and the French regulatory body on data privacy (Commission nationale de l'informatique et des liberté) (Autorisation N˚ 916397 (DR-2016-459).

**Funding:** ComPaRe was funded by the Assistance Publique Hôpitaux de Paris and the Université de Paris. The funders had no role in study design, data collection and analysis, decision to publish, or preparation of the manuscript.

**Competing interests:** The authors have declared that no competing interests exist.

## Conclusion

Individuals with chronic conditions have distorted perceptions of their risk of severe illness with COVID-19. In addition, they are exposed to COVID-19 due to their context or attitudes.

## Introduction

The novel coronavirus disease 2019 (COVID-19) pandemic threatens to saturate healthcare systems all around the world [1]. On April 7[th] of July 2020, 6,416,828 cases were confirmed in 213 countries, with 382,867 deaths [2]. In France, 152,444 cases were confirmed, with 29,065 deaths [3]. Severe acute respiratory distress develops in about 16% to 26% of patients hospitalized with COVID-19, thus requiring oxygen supplementation and/or intensive care [4]. As the number of cases grows worldwide, in order to avoid a surge of demand on the healthcare system and shortages of equipment such as ventilators needed to care for critically ill patients [5–7], many countries have imposed quarantine and recommended physical distancing to reduce transmission to people likely to have a severe illness (i.e., older patients and those with chronic comorbidities). Those individuals with chronic comorbidities should also, in return, avoid contacts and/or use appropriate measures to prevent potential infection. Yet, in France and around the world, specific advice for individuals with chronic conditions and their households is scarce with most information is intended for the general public. For example, information from the European Centre for Disease Prevention and Control refer to only "people with chronic diseases" without specifying specific groups of individuals. This was confirmed by a recent study showing that adults with comorbid conditions lacked critical knowledge about COVID-19 [8].

In this study, we aimed to understand the perceptions, context and attitudes toward COVID-19 of individuals with chronic conditions in order to clarify their potential risk of infection.

## Material and methods

This study was a cross-sectional survey nested in ComPaRe, a nationwide e-cohort of patients with chronic conditions in France [9].

### Participants

Participants were adults with chronic conditions recruited from the Community of Patients for Research (ComPaRe, http://compare.aphp.fr), a nationwide e-cohort of patients with chronic conditions in France. Participants of ComPaRe are adults (>18 years old) who reported having at least one chronic condition (defined as a condition requiring healthcare for at least 6 months) and who joined the project to donate time to accelerate research of their conditions by answering regular patient-reported outcomes and experience measures online [9]. All participants provide electronic informed consent before participating in the e-cohort. ComPaRe was approved by the Comité de Protection des Personnes Ile de France 1 (IRB: 0008367). All methods were performed in accordance with the relevant guidelines and regulations.

### Context and settings

Data from this study were collected between March 23 and April 2, 2020 at the peak of the French epidemic. During that time, 27 475 new cases of COVID-19 were confirmed, with a

total of 56 261 cases on April 2, 2020. This time period includes the maximum number of daily cases in France (April 1, 2020) [10]. Since March 17, France had been under lockdown (movement restrictions and closure of non-essential businesses), and people with chronic conditions were encouraged to stay at home [11]. During this time, knowledge of COVID-19 was still limited and information for the public was imprecise. For example, information available on the website of the French ministry of health referred to "people at risk", mixing older people and patients with chronic conditions [12]. Of note, at the time of the study, benefits of using face masks were debated in France and in Europe.

## Data collected

Participants' demographic and clinical data were collected as part of their participation in the ComPaRe e-cohort. All variables are updated yearly. Conditions and medications are self-reported by patients by using the International Classification of Primary Care-Version 2 [13] and the Thesorimed database of medications (French database of medications developed by the national health insurance) [14].

In addition, participants answered a dedicated survey designed by VTT and PR by use of the literature and their own expertise [8]. It was then face-validated by two other researchers (IP and CR) with expertise in questionnaire development before dissemination. The questionnaire was not tested with patients; however, the first respondents provided comments in a dedicated open-ended question at the end of the questionnaire, which led to minor reformulations. Final survey questions are available in S1 and S2 Data. This survey covered 3 topics.

- For perception of risk of severe illness with COVID-19, we asked participants whether they felt at high risk of severe illness with COVID-19 with the question: "Do you feel at increased risk of severe illness with COVID-19 as compared to people of the same age as you but without chronic disease?" (yes/no).

- For their context. participants described their activity (e.g., whether they continued working outside of the home); their household (i.e., whether any member of their household worked outside of the home and were in contact with the public); and their recent physical visits to healthcare professionals.

- For their attitudes to prevent infection, participants were presented four theoretical situations involving different types of contacts: frequent (e.g., family member frequently visiting, child care, etc.) or occasional (e.g., during shopping) and discerning whether these contacts showed symptoms or not. In each situation, participants reported whether they would refuse contact, enact physical distancing or wear personal protective equipment (mask, gloves, etc.).

## Analysis

Results of the survey were described globally and for the subgroup of patients considered at high risk of a severe illness according to the French High Council for Public Health (Box 1). These patients were those with a severe cardiac or vascular disease (high blood pressure with complications, history of stroke or ischemic heart disease, cardiac surgery, heart failure), insulin-dependent diabetes, chronic lung disease or lung disease likely to be exacerbated by a viral infection, chronic kidney disease under dialysis, cancer under treatment, immunodeficiency (due to a drug [cancer chemotherapy, immunosuppressive medications, biotherapy and/or corticosteroids], an uncontrolled HIV infection, transplantation, or cancer), liver cirrhosis, or severe obesity (body mass index [BMI] >40 kg/m$^2$), or pregnant in the third trimester [15]. To

> **Box 1. Patients at high risk of severe COVID-19 according to the French High Council for Public Health [15].**
>
> **According to the literature**:
>
> - Age $\geq$ 70 years
>
> - History of cardiovascular disease: complicated hypertension, history of stroke or coronary artery disease, heart surgery, heart failure (New York Heart Association class III or IV)
>
> - Insulin-dependent diabetes or with diabetic microangiopathy or macroangiopathy
>
> - Chronic respiratory disease likely to result in decompensation during a viral infection
>
> - Chronic renal failure on dialysis
>
> - Cancer under treatment
>
> **Despite the lack of data in the literature, the following patients are also considered at high risk based on available data on other respiratory infections**:
>
> - Cancer chemotherapy, immunosuppressive therapy, biotherapy and/or immunosuppressive dose corticosteroid therapy (= high-risk treatment in this article)
>
> - Uncontrolled HIV infection or CD4 count <200/mm3, solid organ or hematopoietic stem cell transplant, or blood cancer under treatment
>
> - Cirrhosis at least stage B of the Child-Pugh classification
>
> - Morbid obesity (body mass index > 40 kg/m$^2$) by analogy with influenza A (H1N1)09
>
> - Third trimester of pregnancy

operationalize these criteria with the data available in ComPaRe, one physician (VTT) matched the conditions and treatments reported by patients in ComPaRe with the list of high-risk conditions and treatments presented above. Cancer chemotherapy immunosuppressive medications, biotherapy and corticosteroids were those classified as such in manufacturers' prescribing information, by using the Vidal dictionary (https://www.vidal.fr/classifications/vidal/).

Descriptive statistics (mean with SD and frequency with percentage) were calculated for all patient characteristics and survey responses. Associations between participant characteristics and responses to the survey items were then examined in bivariate analyses by chi-square or *t* test, as appropriate. In addition, we fitted two logistic regressions aimed at exploring the association between participants' characteristics and 1) their perception of their risk for severe infection and 2) their attitudes to prevent infection with occasional contacts with asymptomatic people. Variables included in the model were sex, age (as a continuous variable), household with > 1 person (including the patient), low educational level, smoking status (current smoker vs. others), treatment considered at risk according to the French High Council for Public Health, BMI $\geq$40 kg/m$^2$, high blood pressure, diabetes (under insulin treatment or not), history of stroke or cardiac ischemic disease, heart failure (any New York Heart Association stage), asthma, chronic obstructive pulmonary disease, thyroid disease, chronic kidney failure (under dialysis or not), cancer (under treatment or not) and osteoarthritis. Analyses were performed on complete cases only. P < 0.05 was considered statistically significant. No corrections for multiple testing were performed.

Analyses involved using a weighted dataset obtained by calibration on margins with weights for age categories (<24, 25–34, 35–44, 45–54, 55–64, 65–74, >75 years), sex and educational level (low, middle school or equivalent, high school or equivalent, associate's degree, higher education). Weights were derived from national census data describing the French population reporting chronic conditions [16, 17].

Analyses involved use of R v3.6.1 (http://www.R-project.org, the R Foundation for Statistical Computing, Vienna, Austria).

## Results

### Participants

Between March 23 and April 2, 2020, we invited 18,651 patients from ComPaRe to complete our survey and 7169 (38.4%) answered (S1 Fig). Participants were mostly female (5616 [78.3%]) with mean (SD) age 46.1 (14.7) years. In the non-weighted data, diseases most frequently reported were high blood pressure (11.6%), diabetes (7.1%), asthma (6.2%) and cancer (5.2%); 3684 (51.4%) participants reported ≥2 chronic conditions. Differences between respondents and non-respondents are shown in S1 Table. In the weighted sample, 39.4% were at high risk for a severe illness according to the French recommendations: 33.0% because of their conditions, 8.8% because of their treatments, 1.9% with BMI > 40 kg/m$^2$, and 0.5% in their third trimester of pregnancy. Patients' characteristics before and after weighting are presented in Table 1.

### Perception of the risk of severe illness with COVID-19

In the weighted sample, 63% of participants felt at risk of severe illness with COVID-19, of whom 51% (32% of the whole sample) reported a high-risk situation according to the French High Council for Public Health. Conversely, 37% participants did not feel at risk of severe COVID-19, of whom 20% (7.4% of the whole sample) reported a high-risk situation according to the French High Council for Public Health (Fig 1). Patients' characteristics associated with a perceived risk of severe COVID-19 identified in the logistic regressions are presented in Table 2.

### Potential risk of infection due to context

In total, 7041 (98%) participants answered the survey section regarding their risk of infection due to their context. Risk of infection involved working outside of the home (8.8% of participants, of whom 29% were care professionals), visits to health facilities for a consultation or test (54.7% of participants) or to a pharmacy (82% of participants); their household (74.9% of participants lived with at least on other person, of whom 18% worked outside of the home and 13% were children < 15 years old), or regular contacts with people outside of their home (e.g., family, friends, housekeeping, child care, etc.) (5% of participants). In all, 23.7% were exposed to some risks because of their work, their household members working outside of the home, or regular visits from external contacts.

Among patients at high risk of a severe illness according to the French High Council for Public Health, 5% continued working, 15% had a household member working outside of the home and 7% reported regular contacts with people outside of their home. In all, 21.1% were exposed to some risks because of their work, their household members working outside of the home, or regular visits from external contacts.

**Table 1. Participant characteristics (n = 7169).**

| Characteristic | Raw dataset (n = 7169) | Weighted dataset[1] (n = 7169) |
|---|---|---|
| **Age, mean (SD)–yr** | 46.1 (14.7) | 55.1 (17.0) |
| **Female sex—no (%)** | 5616 (78.3) | 3788 (52.8) |
| **Educational level—no (%)** | | |
| Low | 386 (5.4) | 699 (9.8) |
| Middle school or equivalent | 1164 (16.2) | 4039 (56.3) |
| High school or equivalent | 533 (7.4) | 991 (13.8) |
| Associate's degree | 1510 (21.1) | 629 (8.8) |
| Higher education | 3576 (49.9) | 811 (11.3) |
| **Smoking status** | | |
| Never smoker | 2319 (46.3) | 2554 (35.6) |
| Former smoker | 1671 (33.3) | 3371 (47) |
| Current smoker (occasional) | 290 (5.8) | 309 (4.3) |
| Current smoker (frequent) | 732 (14.6) | 926 (12.9) |
| Missing | 2 (0.03) | 9 (0.1) |
| **Multimorbid—no (%)** | 3684 (51.4) | 4065 (56.7) |
| **Number of diseases, mean (SD)** | 2.3 (2.2) | 2.5 (2.4) |
| **Conditions[2]—no (%)** | | |
| High blood pressure | 834 (11.6) | 1486 (20.7) |
| Diabetes | 506 (7.1) | 819 (11.4) |
| Stroke or cardiac ischemic disease | 70 (1.0) | 109 (1.5) |
| Heart failure (other than ischemic diseases) | 79 (1.1) | 91 (1.3) |
| Asthma | 448 (6.2) | 390 (5.4) |
| COPD | 124 (1.7) | 260 (3.6) |
| Thyroid disease | 362 (5.0) | 294 (4.1) |
| Chronic kidney failure | 142 (2.0) | 276 (3.8) |
| Cancer | 373 (5.2) | 515 (7.2) |
| Osteoarthritis | 319 (4.4) | 388 (5.4) |
| Inflammatory rheumatic diseases | 407 (5.7) | 432 (6.0) |
| **High-risk situation according to the French High Council for Public Health—no (%)** | 2152 (30.0) | 2828 (39.4) |
| High-risk conditions[3] | 1683 (23.5) | 2367 (33.0) |
| High-risk treatments[3] | 513 (7.2) | 628 (8.8) |
| Third trimester of pregnancy | 81 (1.1) | 33 (0.5) |
| BMI $\geq$ 40 kg/m$^2$ | 124 (1.7) | 139 (1.9) |

COPD, chronic obstructive pulmonary disease; BMI, body mass index

[1] Weighted data were obtained after calibration on margins for sex, age categories and educational level by using data from a national census describing the French population self-reporting at least one chronic condition.

[2] A patient may have multiple chronic conditions.

[3] High-risk conditions and treatment are according to the French High Council for Public Health

## Potential risk of infection due to attitudes

In total, 6940 (97%) participants answered the survey section regarding their attitudes to prevent infections. Independent of the type of contact, participants reported that they would enact physical distancing under all situations presented to them. About one quarter of patients would refuse any contact with symptomatic people (17.8% and 23.4% for occasional and

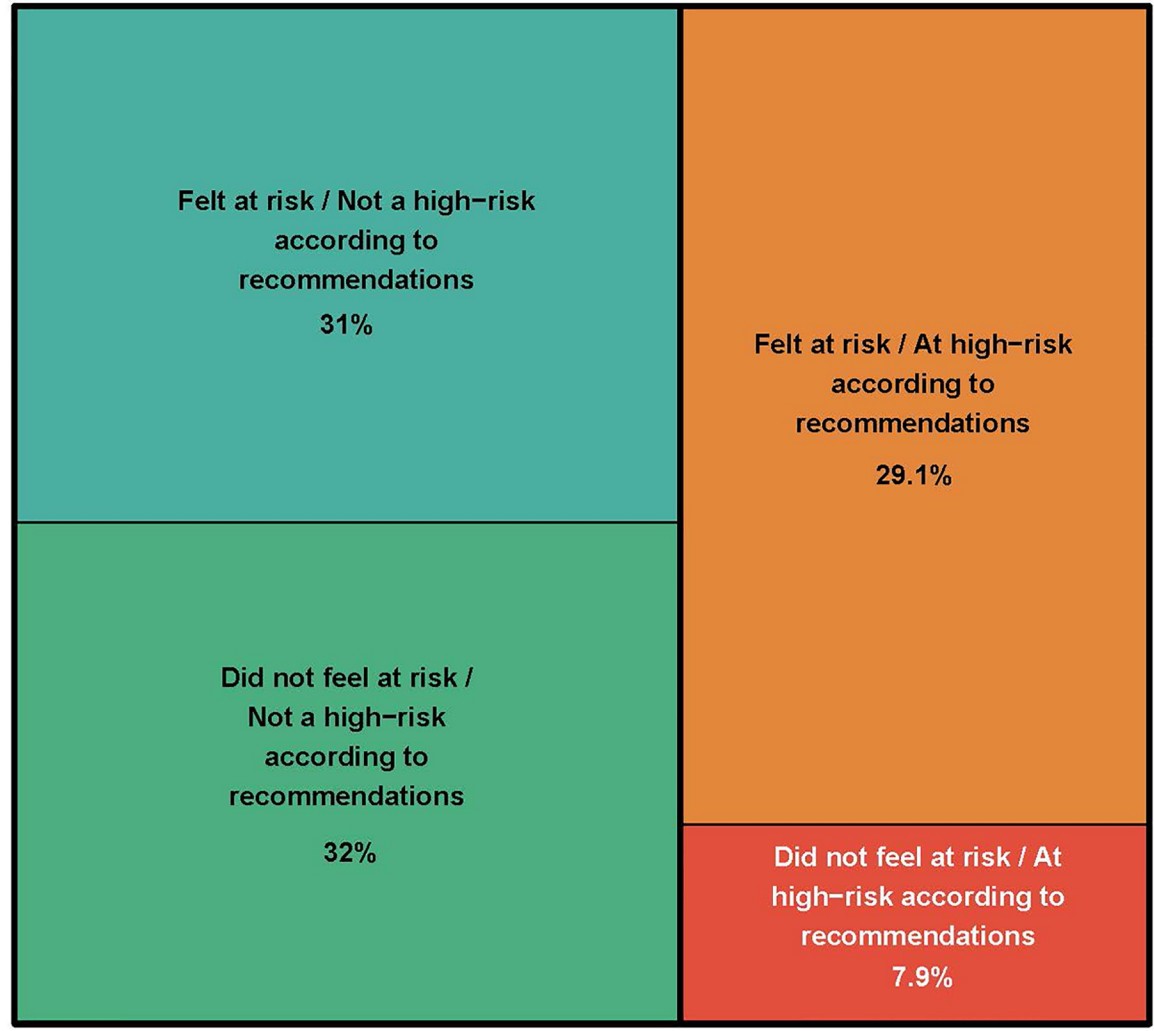

**Fig 1. Participants' perception of the risk of severe illness with COVID-19 depending on their actual risk of severe illness according to recommendations from the French High Council for Public Health (n = 7169).** Surface is proportional to the number of patients in each category in the weighted analysis (calibration on margins for sex, age categories and educational level by using data from a national census describing the French population self-reporting at least one chronic condition).

frequent contacts, respectively). Concerning the use of personal protective equipment, use of masks ranged from 19% (for occasional contacts with asymptomatic people) to 65% (for frequent contacts with symptomatic people). Similarly, use of gloves ranged from 19% (for occasional contacts with asymptomatic people) to 50% (for frequent contacts with symptomatic people) (Fig 2).

We found similar results in the subgroup of patients at high risk of a severe illness according to the French High Council for Public Health. Only 18.2% and 23.2% patients would refuse contact with symptomatic people for occasional and frequent contacts, respectively. Concerning the use of personal protective equipment, use of masks ranged from 30% (for occasional contacts with asymptomatic people) to 63% (for frequent contacts with symptomatic people).

**Table 2. Association between participant characteristics and their perceived risk of severe COVID-19.** Results of logistic regression analysis of complete cases, accounting for weights obtained after calibration on margins for sex, age categories and educational level by using data from a national census describing the French population self-reporting at least one chronic condition.

| Characteristic | Odds ratio (95% confidence interval) |
| --- | --- |
| Female sex | 1.33 (1.04–1.71)* |
| Age | 1.01 (1–1.02)* |
| Current smoker | 1.21 (0.9–1.62) |
| Household > 1 person | 1.12 (0.83–1.52) |
| Low educational level | 1.1 (0.76–1.59) |
| High risk treatments | 14.08 (6.16–32.18)* |
| BMI $\geq$ 40 kg/m$^2$ | 1.77 (0.81–3.87) |
| High blood pressure | 1.13 (0.75–1.69) |
| Diabetes | 3.02 (1.83–4.98)* |
| Stroke or cardiac ischemic disease | 9.16 (2.26–37.19)* |
| Heart failure | 2.4 (0.67–8.63) |
| Asthma | 4.64 (2.22–9.68)* |
| COPD | 6.32 (2.14–18.73)* |
| Thyroid disease | 1.05 (0.55–1.97) |
| Chronic kidney disease | 2.64 (1.01–6.89)* |
| Cancer | 1.93 (0.99–3.73) |
| Osteoarthritis | 0.98 (0.59–1.62) |

* $P < 0.05$

BMI, body mass index; COPD, chronic obstructive pulmonary disease

Similarly, use of gloves ranged from 21% (for occasional contacts with asymptomatic people) to 44% (for frequent contacts with symptomatic people).

Patients' characteristics associated with a perceived risk of severe illness with COVID-19 identified on logistic regression are presented in Table 3. The only variable found associated with use of face masks with asymptomatic people (or refusal to see these people) was patients' perception of high risk of severe infection by COVID-19 (odds ratio 1.93, 95% confidence interval 1.53–2.43).

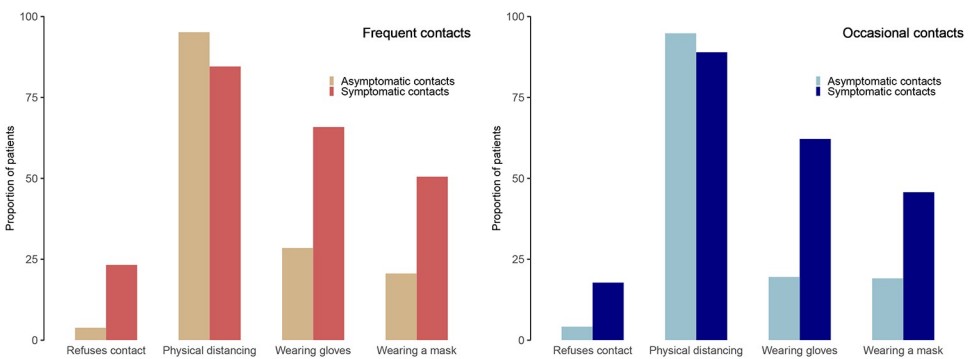

**Fig 2. Participant-reported attitudes to prevent infection in situations involving frequent or occasional contacts with symptomatic or asymptomatic people (n = 6940).**

**Table 3. Association between participant characteristics and the use of face masks for occasional contacts with asymptomatic people (or the refusal to see these people).** Results of logistic regression analysis of complete cases, accounting for weights obtained after calibration on margins for sex, age categories and educational level by using data from a national census describing the French population self-reporting at least one chronic condition.

| Characteristic | Odds ratio (95% confidence interval) |
|---|---|
| Female sex | 1.18 (0.92–1.5) |
| Age | 1 (0.99–1.01) |
| Current smoker | 0.94 (0.69–1.28) |
| Household > 1 person | 1.2 (0.93–1.56) |
| Low educational level | 0.95 (0.67–1.34) |
| High-risk treatments | 1 (0.65–1.53) |
| BMI $\geq$ 40 kg/m$^2$ | 1.31 (0.73–2.35) |
| High blood pressure | 0.73 (0.52–1.03) |
| Diabetes | 1.06 (0.72–1.55) |
| Stroke or cardiac ischemic disease | 1.04 (0.44–2.48) |
| Heart failure | 2.36 (0.9–6.23) |
| Asthma | 0.85 (0.59–1.23) |
| COPD | 0.89 (0.42–1.87) |
| Thyroid disease | 1.25 (0.79–1.97) |
| Chronic kidney disease | 0.83 (0.4–1.73) |
| Cancer | 0.66 (0.43–1.03) |
| Osteoarthritis | 1.42 (0.91–2.23) |
| Feeling at risk of severe COVID-19 | 1.93 (1.53–2.44) |

* $P < 0.05$

## Discussion

We involved 7169 individuals with chronic conditions in a nationwide survey nested in an existing cohort and described their perception of risk of severe COVID-19 and their potential risk of infection due to context and attitudes.

First, our study highlighted that patients with chronic conditions have distorted perceptions of their risk of severe COVID-19. Among patients with criteria for high risk of severe COVID-19 by the French High Council for Public Health (40% of our sample), about 20% did not feel at risk and could therefore adopt incautious attitudes. This figure may even be conservative in light of recent works suggesting that all patients with hypertension, diabetes, cardiovascular disease, or chronic lung disease are at risk, not just those with complicated diseases [18–20]. Data from the he Chinese Center for Disease Control and Prevention showed increased case fatality rate among patients with preexisting comorbid conditions—10.5% for cardiovascular disease, 7.3% for diabetes, 6.3% for chronic respiratory disease, 6.0% for hypertension, and 5.6% for cancer [20]. Especially, our findings highlight that patients with a BMI $\geq$ 40 kg/m$^2$ or who smoked did not feel at risk nor took extra precautions when in contact with other people despite these two factors being associated with risk of severe complications and mortality from COVID-19 [21, 22].

These results are of importance because of the confluence of two elements. First, preventing infection for people at risk of severe disease is difficult. In our study, 21.2% of patients at high risk of a severe illness according to French recommendations were in frequent contact with "the outside world" during the quarantine because of their work, their household members working outside of the home, or regular visits from external contacts. Second, feeling at risk seems to be the major factor for using face masks with asymptomatic people. At the time of the

study, it was still unknown that 40% to 80% of transmission events could occur from people who are presymptomatic or asymptomatic [23]. Therefore, specific communication clearly identifying patients at risk for severe illness by COVID-19 is mandatory. Communication should also target the household of these patients because the rate of secondary transmission among household contacts of patients with SARS-CoV-2 infection was estimated at 30% [24].

Our results are important given the cumulative amount of evidence showing that patients with chronic conditions, about 20 million individuals in France, are at increased risk of severe COVID and death. In a small case series conducted at the beginning of the epidemic in China, among 102 patients hospitalized for COVID-19, those with comorbidities (especially hypertension, diabetes, cardiovascular and respiratory diseases) were more likely to be hospitalized in Intensive care units [25, 26]. Similar findings were observed in Europe. In a large case series of 4000 patients hospitalized in ICUs in Italy, the highest risk of death was for patients with chronic obstructive pulmonary disease (adjusted HR [aHR] 1.68, 95% CI 1.28–2.19) and type 2 diabetes (aHR 1.18, 95% CI 1.01–1.39) [27]. Reasons underlying these findings are still unclear, with hypotheses related to meta-inflammation or use of angiotensin-converting enzyme inhibitors (ACEIs)/angiotensin receptor blockers (ARBs) in these populations, despite recent controversial findings about this latter point [28, 29].

Our study complements the literature on the awareness and attitudes of patients with chronic conditions related to COVID-19. To date, most works have focused on the general public [30, 31]. Knowledge and attitudes of patients with chronic conditions is unknown, apart from a study of 600 patients with chronic conditions in the United States that showed gaps in awareness and knowledge of COVID-19 among patients with chronic conditions [8]. Our findings confirm these results and provide details on individuals' risks associated with their context and their attitudes to prevent infection.

This study has several limitations. First, all data were self-reported, with risk of desirability bias regarding their attitudes. Second, individuals at high risk of severe illness with COVID-19 are not yet well known; recommendations from the French High Council for Public Health are mostly based on case reports in China and precaution measures [15]. Third, the response rate was relatively low (38%) owing to the short duration of data collection (10 days) and the sole use of e-mails for the invitation and reminders. Yet, such response rate is consistent with the literature of online surveys for the general public [32, 33]. Non-respondents were younger, less multimorbid and had fewer conditions considered at high risk according to recommendations than respondents. Despite statistical weighting, results should be generalized with caution.

In conclusion, we found that individuals with chronic conditions may have distorted perceptions of their risk of severe illness with COVID-19. Targeted communication may increase the use of personal protective equipment and prevent infection, which is fundamental because 20% of these individuals are exposed to infection because of their work, their household or regular visits from external contacts, despite quarantine.

## Supporting information

**S1 Data. Questionnaire for participants (French).**
(DOCX)

**S2 Data. Questionnaire for participants (English).**
(DOCX)

**S1 Fig. Flow chart of participants' answers to the survey.**
(DOCX)

**S1 Table. Demographic characteristics of respondents and non-respondents to the survey (raw data).**
(DOCX)

# Acknowledgments

The authors thank Isabelle Pane and Carolina Riveros for their help in the survey development and Isabelle Pane for data management.

# Author Contributions

**Conceptualization:** Viet-Thi Tran, Philippe Ravaud.

**Data curation:** Viet-Thi Tran.

**Formal analysis:** Viet-Thi Tran.

**Methodology:** Viet-Thi Tran, Philippe Ravaud.

**Resources:** Philippe Ravaud.

**Supervision:** Philippe Ravaud.

**Writing – original draft:** Viet-Thi Tran.

**Writing – review & editing:** Viet-Thi Tran, Philippe Ravaud.

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
