## [Decision Letter · Decision Letter 0]

29 May 2020

PONE-D-20-14011

Potential risk of COVID-19 in adults with chronic conditions because of their perceptions and attitudes: results from a cross-sectional survey nested in the ComPaRe e-cohort

PLOS ONE

Dear Dr. Tran,

Thank you for submitting your manuscript to PLOS ONE. After careful consideration, we feel that it has merit but does not fully meet PLOS ONE’s publication criteria as it currently stands. Therefore, we invite you to submit a revised version of the manuscript that addresses the points raised during the review process.

We look forward to receiving your revised manuscript.

Kind regards,

Wen-Jun Tu

Academic Editor

PLOS ONE

Journal Requirements:

2. Please include additional information regarding the survey or questionnaire used in the study and ensure that you have provided sufficient details that others could replicate the analyses. For instance, if you developed a questionnaire as part of this study and it is not under a copyright more restrictive than CC-BY, please include a copy, in both the original language and English, as Supporting Information."

3. Please state whether you validated the questionnaire prior to testing on study participants. Please provide details regarding the validation group within the methods section.

5. . Thank you for stating the following financial disclosure:

The funders had no role in study design, data collection and analysis, decision to

publish, or preparation of the manuscript.

Reviewers' comments:

Reviewer's Responses to Questions

**Comments to the Author**

1. Is the manuscript technically sound, and do the data support the conclusions?

Reviewer #1: Partly

2. Has the statistical analysis been performed appropriately and rigorously? 

Reviewer #1: No

3. Have the authors made all data underlying the findings in their manuscript fully available?

Reviewer #1: No

4. Is the manuscript presented in an intelligible fashion and written in standard English?

Reviewer #1: No

5. Review Comments to the Author

Reviewer #1: This is peer-review of paper by Viet-Thi Tran et al, which determined Potential risk of COVID-19 in adults with chronic conditions because of their perceptions and attitudes in France. They found that those with chronic medical illnesses have skewed views about their likelihood about severe disease with COVID-19.

The strengths of this study are the large sample size.

Major Comments:

1. The survey design was omitted for us as evaluators, who designed it? How? Is it validated? Elaborate more to this. It should be presented in the appendix.

2. The response rate was very low, what was the reason? How did you distribute it?

3. The analysis was mainly descriptive with no interesting associations came out from this survey. Make it more interesting for the readers and researchers.

4. The definition of the high risk of severe illness is not clear and misleading.

5. Why did you divide Perception of risk of severe illness with COVID-19 to four categories? Explain further and make a valid point.

6. Your survey was looking for perception of people but attitude of people was presented and analysed, please be specific about your objectives and don’t confuse the readers.

7. Very limited discussion was presented; no scientific discussion has been done which very important papers were not discussed concerning prevalence, severity and mortality of chronic respiratory conditions, cardiac, renal, and other comorbidities.

6. PLOS authors have the option to publish the peer review history of their article (what does this mean?). If published, this will include your full peer review and any attached files.

Reviewer #1: No

---

## [Author Response · Author response to Decision Letter 0]

26 Jun 2020

Dear Editor,

Thank you very much for giving us the possibility to answer the main concerns raised by the two reviewers of our manuscript entitled “COVID-19–related perceptions, context and attitudes of adults with chronic conditions: results from a cross-sectional survey nested in the ComPaRe e-cohort” (PONE-D-20-14011). You'll find attached the detailed point by point answers to reviewer and editor comments.

---

## [Decision Letter · Decision Letter 1]

6 Jul 2020

PONE-D-20-14011R1

COVID-19–related perceptions, context and attitudes of adults with chronic conditions: results from a cross-sectional survey nested in the ComPaRe e-cohort

PLOS ONE

Dear Dr. Tran,

Thank you for submitting your manuscript to PLOS ONE. After careful consideration, we feel that it has merit but does not fully meet PLOS ONE’s publication criteria as it currently stands. Therefore, we invite you to submit a revised version of the manuscript that addresses the points raised during the review process.

We look forward to receiving your revised manuscript.

Kind regards,

Wen-Jun Tu

Academic Editor

PLOS ONE

Reviewers' comments:

Reviewer's Responses to Questions

**Comments to the Author**

1. If the authors have adequately addressed your comments raised in a previous round of review and you feel that this manuscript is now acceptable for publication, you may indicate that here to bypass the “Comments to the Author” section, enter your conflict of interest statement in the “Confidential to Editor” section, and submit your "Accept" recommendation.

Reviewer #1: (No Response)

Reviewer #2: (No Response)

2. Is the manuscript technically sound, and do the data support the conclusions?

Reviewer #1: Yes

Reviewer #2: (No Response)

3. Has the statistical analysis been performed appropriately and rigorously? 

Reviewer #1: Yes

Reviewer #2: (No Response)

4. Have the authors made all data underlying the findings in their manuscript fully available?

Reviewer #1: Yes

Reviewer #2: (No Response)

5. Is the manuscript presented in an intelligible fashion and written in standard English?

Reviewer #1: Yes

Reviewer #2: (No Response)

6. Review Comments to the Author

Reviewer #1: Thank you for the authors for addressing all the comments except ONE, which is number (7. A very limited discussion was presented; no scientific discussion has been done which very important papers were not discussed concerning prevalence, severity and mortality of chronic respiratory conditions, cardiac, renal, and other comorbidities). They need to cite systematic reviews and meta-analysis about the above comorbidities and link that with the current prevalence and severity reported in France. The other important point here is about Attitude, I can not see anything about smoking habits and the risk of getting an infection with those vulnerable group of patients. A recent meta-analysis published by Plos One highlighted the prevalence and severity associated with former, never and current smokers and COVID-19. Please discuss this in your discussion and make a comparison with your results as this definitely ATTITUDE.

Good luck

Reviewer #2: The authors studied the perceptions, context and attitudes of individuals with chronic conditions during the COVID-19 pandemic to clarify their potential risk of infection,the result was that 63% patients felt at risk because of severe illness, about 23.7% were at risk of infection, at the end they concluded that individuals with chronic conditions have distorted perceptions of their risk of severe illness with COVID-19.So I will give some comments as followings.

In this study, the investigation time period was selected from 3.23-4.2. I want to konw the reason for choosing this study time? Why not choosing a longer period of time or another time period?

In this study, how to define a serious disease state? What types of diseases are included? Whether different disease types will affect the conclusions of this investigation

In the face of the COVID-19 epidemic, what are the current policy measures adopted by the French government?

The following references should be discussed in the revision text.

Cao JL, Hu XR, Tu WJ., & Liu Q. (2020). Clinical Features and Short-term Outcomes of 18 Patients with Corona Virus Disease 2019 in Intensive Care Unit. Intensive Care Medicine, DOI: 10.1007/s00134-020- 05987-7.

Cao JL, Tu WJ, Hu XR, & Liu Q. (2020). Clinical Features and Short-term Outcomes of 102 Patients with Corona Virus Disease 2019 in Wuhan,China. Clinical Infectious Diseases,DOI: 10.1093/cid/ciaa243/ 5814897.

7. PLOS authors have the option to publish the peer review history of their article (what does this mean?). If published, this will include your full peer review and any attached files.

Reviewer #1: No

Reviewer #2: No

---

## [Author Response · Author response to Decision Letter 1]

22 Jul 2020

Dear Editor,

Thank you for giving us the possibility to answer the main concerns raised by the two reviewers of our manuscript entitled “COVID-19–related perceptions, context and attitudes of adults with chronic conditions: results from a cross-sectional survey nested in the ComPaRe e-cohort” (PONE-D-20-14011R1).

Reviewers mainly asked for changes in the discussion to detail the link between COVID-19 infection and chronic conditions (e.g. discussing the prevalence, severity and mortality of chronic respiratory conditions, cardiac, renal, and other comorbidities in France). These are, of course, topics of critical importance but we are not sure that they are within the scope of the paper, which was to explore patients’ perceptions and attitudes toward the risk of COVID-19. As requested by the reviewers, we added more details in the discussion to provide a basis for reflection, but we believe that expanding this section too much may lead readers astray.

Reviewer #1: Thank you for the authors for addressing all the comments except ONE, which is number (7. A very limited discussion was presented; no scientific discussion has been done which very important papers were not discussed concerning prevalence, severity and mortality of chronic respiratory conditions, cardiac, renal, and other comorbidities). They need to cite systematic reviews and meta-analysis about the above comorbidities and link that with the current prevalence and severity reported in France. 

We added some details to discuss the importance of understanding the perceptions of patients with chronic conditions toward COVID-19. 

Our results are important given the cumulative amount of evidence showing that patients with chronic conditions, about 20 million individuals in France, are at increased risk of severe COVID and death. In a small case series conducted at the beginning of the epidemic in China, among 102 patients hospitalized for COVID-19, those with comorbidities (especially hypertension, diabetes, cardiovascular and respiratory diseases) were more likely to be hospitalized in Intensive care units. Similar findings were observed in Europe. In a large case series of 4000 patients hospitalized in ICUs in Italy, the highest risk of death was for patients with chronic obstructive pulmonary disease (adjusted HR [aHR] 1.68, 95% CI 1.28-2.19) and type 2 diabetes (aHR 1.18, 95% CI 1.01-1.39). Reasons underlying these findings are still unclear, with hypotheses related to meta-inflammation or use of angiotensin-converting enzyme inhibitors (ACEIs)/angiotensin receptor blockers (ARBs) in these populations, despite recent controversial findings about this latter point.

The other important point here is about Attitude, I cannot see anything about smoking habits and the risk of getting an infection with those vulnerable group of patients. A recent meta-analysis published by Plos One highlighted the prevalence and severity associated with former, never and current smokers and COVID-19. Please discuss this in your discussion and make a comparison with your results as this definitely ATTITUDE.

Our objective was to assess patients’ perceptions regarding COVID-19 and their attitudes regarding protection measures. We did not collect whether patients stopped smoking because of the fear of severe COVID. 

However, we added details about patients’ current smoking status and their perception of risk for severe COVID-19. Especially, we now describe in table 1 patients’ smoking status and we added patients’ smoking status in the two models, which did not change the results. Finally, we discussed the fact that patients with BMI ≥ 40 kg/m2 or who smoked did not feel at risk and did not take specific measures to protect themselves despite these two factors being associated with risk of severe complications and mortality from COVID-19.

Especially, our findings highlight that patients with a BMI ≥ 40 kg/m² or who smoked did not feel at risk nor took extra precautions when in contact with other people despite these two factors being associated with risk of severe complications and mortality from COVID-19

Reviewer #2: The authors studied the perceptions, context and attitudes of individuals with chronic conditions during the COVID-19 pandemic to clarify their potential risk of infection, the result was that 63% patients felt at risk because of severe illness, about 23.7% were at risk of infection, at the end they concluded that individuals with chronic conditions have distorted perceptions of their risk of severe illness with COVID-19.So I will give some comments as followings.

In this study, the investigation time period was selected from 3.23-4.2. I want to know the reason for choosing this study time? Why not choosing a longer period of time or another time period?

The study was conducted at the peak of the epidemic in France and our results were intended to provide a quick insight into the perceptions and attitudes of chronic patients toward the risk for severe infection. 

We now report in the settings section, the number of confirmed cases at the time of the study.

Data from this study were collected between March 23 and April 2, 2020 at the peak of the French epidemic. During that time, 27 475 new cases of COVID-19 were confirmed, with a total of 56 261 cases on April 2, 2020. This time period includes the maximum number of daily cases in France (April 1, 2020).

Because patients are recruited from an existing cohort, so a longer response period would have, at best, increased the survey response rate. We already describe the difference between respondents and non-respondents and discuss the response rate in comparison to other surveys conducted at the same time. 

In this study, how to define a serious disease state? What types of diseases are included? 

Any participant with a chronic condition can participate in ComPaRe. Patients self-report their chronic conditions by using a list of 217 diseases, inspired from the International Classification of Primary Care-Version 2. 

Details on the ComPaRe e-cohort have been published elsewhere (Tran VT, J Clin Epidemiol 2020). The full protocol of the ComPaRe cohort is available on the cohort website www.compare.aphp.fr.

Whether different disease types will affect the conclusions of this investigation

Associations identified in this study were independent of the presence of several conditions (high blood pressure, diabetes, stroke, ischemic heart failure, asthma, COPD, thyroid disease, chronic kidney disease, cancer and osteoarthritis). Indeed, these diseases were included as predictors in the non-parsimonious logistic regression analysis exploring the association between patient characteristics, their perception of their risk for severe infection (Table 2) and their use of face masks for occasional contacts with asymptomatic people (Table 3).

In the face of the COVID-19 epidemic, what are the current policy measures adopted by the French government?

At the time of the study, France had been under lockdown (movement restrictions and closure of non-essential businesses) and people with chronic conditions were encouraged to stay at home. Yet public communication for people with chronic conditions was imprecise. In particular, there was no consensus on the use of face-masks.

Since March 17, France had been under lockdown (movement restrictions and closure of non-essential businesses), and people with chronic conditions were encouraged to stay at home. During this time, knowledge of COVID-19 was still limited and information for the public was imprecise. For example, information available on the website of the French ministry of health referred to “people at risk”, mixing older people and patients with chronic conditions.

The following references should be discussed in the revision text.

Cao JL, Hu XR, Tu WJ., & Liu Q. (2020). Clinical Features and Short-term Outcomes of 18 Patients with Corona Virus Disease 2019 in Intensive Care Unit. Intensive Care Medicine, DOI: 10.1007/s00134-020- 05987-7.

Cao JL, Tu WJ, Hu XR, & Liu Q. (2020). Clinical Features and Short-term Outcomes of 102 Patients with Corona Virus Disease 2019 in Wuhan,China. Clinical Infectious Diseases,DOI: 10.1093/cid/ciaa243/ 5814897.

We discuss the aforementioned references in the added paragraph in the discussion.

Our results are important given the cumulative amount of evidence showing that patients with chronic conditions, about 20 million individuals in France, are at increased risk of severe COVID and death. In a small case series conducted at the beginning of the epidemic in China, among 102 patients hospitalized for COVID-19, those with comorbidities (especially hypertension, diabetes, cardiovascular and respiratory diseases) were more likely to be hospitalized in Intensive care units. Similar findings were observed in Europe. In a large case series of 4000 patients hospitalized in ICUs in Italy, the highest risk of death was for patients with chronic obstructive pulmonary disease (adjusted HR [aHR] 1.68, 95% CI 1.28-2.19) and type 2 diabetes (aHR 1.18, 95% CI 1.01-1.39). Reasons underlying these findings are still unclear, with hypotheses related to meta-inflammation or use of angiotensin-converting enzyme inhibitors (ACEIs)/angiotensin receptor blockers (ARBs) in these populations, despite recent controversial findings about this latter point.

---

## [Editor Report · Decision Letter 2]

24 Jul 2020

COVID-19–related perceptions, context and attitudes of adults with chronic conditions: results from a cross-sectional survey nested in the ComPaRe e-cohort

PONE-D-20-14011R2

Dear Dr. Tran,

We’re pleased to inform you that your manuscript has been judged scientifically suitable for publication and will be formally accepted for publication once it meets all outstanding technical requirements.

Kind regards,

Wen-Jun Tu

Academic Editor

PLOS ONE
---

## [Editor Report · Acceptance letter]

29 Jul 2020

PONE-D-20-14011R2 

COVID-19–related perceptions, context and attitudes of adults with chronic conditions: results from a cross-sectional survey nested in the ComPaRe e-cohort 

Dear Dr. Tran:

I'm pleased to inform you that your manuscript has been deemed suitable for publication in PLOS ONE. Congratulations! Your manuscript is now with our production department. 

Kind regards, 

on behalf of

Dr. Wen-Jun Tu 

Academic Editor

PLOS ONE